# Bayesian Inverse Modelling for Probabilistic Multi-Nuclide Source Term Estimation Using Observations of Air Concentration and Gamma Dose Rate

**Kasper Skjold Tølløse** [1,2,*] and **Jens Havskov Sørensen** [1]

1   Danish Meteorological Institute, DK-2100 Copenhagen, Denmark
2   Niels Bohr Institute, University of Copenhagen, DK-2100 Copenhagen, Denmark
*   Correspondence: ktoe@dmi.dk

**Abstract:** In case of a release of hazardous radioactive matter to the atmosphere from e.g., a nuclear power plant accident, atmospheric dispersion models are used to predict the spatial distribution of radioactive particles and gasses. However, at the early stages of an accident, only limited information about the release may be available. Thus, there is a need for source term estimation methods suitable for operational use shortly after an accident. We have developed a Bayesian inverse method for estimating the multi-nuclide source term describing a radioactive release from a nuclear power plant. The method provides a probabilistic source term estimate based on the early available observations of air concentration and gamma dose rate by monitoring systems. The method is intended for operational use in case of a nuclear accident, where no reliable source term estimate exists. We demonstrate how the probabilistic formulation can be used to provide estimates of the released amounts of each radionuclide as well as estimates of future gamma dose rates. The method is applied to an artificial case of a radioactive release from the Loviisa nuclear power plant in southern Finland, considering the most important dose-contributing nuclides. The case demonstrates that only limited air concentration measurement data may be available shortly after the release, and that to a large degree one will have to rely on gamma dose rate observations from a frequently reporting denser monitoring network. Further, we demonstrate that information about the core inventory of the nuclear power plant can be used to constrain the release rates of certain radionuclides, thereby decreasing the number of free parameters of the source term.

**Keywords:** source characterization; atmospheric dispersion modelling; inverse modelling; Bayesian inference

## 1. Introduction

In case of a nuclear accident, radioactive particles and gasses may be released to the atmosphere. Consequently, an important part of emergency preparedness is to run simulations with atmospheric dispersion models, thereby predicting the atmospheric distribution as well as deposition of radioactive particles and gasses on the surface of the Earth. However, such models are subject to a number of uncertainties, the most important being the uncertainties of the meteorological predictions, inaccurate physics parameterizations in the dispersion model, and uncertainties of the estimated source term. Immediately after an accident in a nuclear power plant, only limited information about the release may be available. Thus, at the early stages of the accident, the dominating source of uncertainty is most likely the source term. If this is the case, inverse modelling can be used to obtain a source term estimate, which in turn can be used for running the atmospheric dispersion model. The aim of this study is to develop an inverse method for source term estimation, which is suited for operational use for emergency preparedness at the early stages of an accident, i.e., providing a source term estimate based on the limited data available shortly after the accident.

In the early phase of a nuclear power plant accident, a limited number of air concentration observations will be available, and these will typically have a low spatial and temporal resolution, e.g., the filters in such measurement stations may be changed every 24 h or even less frequently. In addition, there may exist gamma dose rate observations at much higher resolution, both spatially and temporally. However, since such measurements are the sum of contributions from all the different radionuclides, it is not clear a priori if they are useful for source term estimation.

Previous studies have used inverse methods for source term estimation. Lately, the still unaccounted for release of Ru-106 in the fall of 2017, was subject to several studies, e.g., [1–4]. However, since the release location has still not been confirmed, the main focus of these studies is localization of the source. The Fukushima Daiichi nuclear disaster in 2011, on the other hand, demonstrated that in-plant monitoring systems may not be working during a severe accident. Thus, different inverse methods have been applied in order to assess the source term. Some studies have estimated the release of certain radionuclides based solely on air concentration measurements [5,6], other include surface deposition measurements [7,8], while other again also include gamma dose rates [9]. Saunier et al. [9] demonstrate that information about ratios between the amounts of certain radionuclides can be used to further constrain the release rates. They use a variational approach to assess the source term, thereby providing a deterministic estimate. However, by using different Bayesian approaches, Liu et al. [6] show that significant uncertainties are associated with the estimated source term, indicating that probabilistic methods are better suited for this type of problem.

Most previous studies in this field aim at estimating the source term associated with accidents a long time after they occurred. However, for emergency preparedness, it is also important to be able to estimate source terms during the early stages, where especially air concentration measurement data are limited. This was addressed by Saunier et al. [9], who further developed their method to be applicable in real-time in case of an accident [10]. Our method is inspired by Saunier et al. [9,10], but instead we use a Bayesian inference method to be able to realistically account for uncertainties of the estimated source term, similar to Liu et al. [6].

The method is applied to an idealized artificial release case from the Finnish Loviisa nuclear power plant. A set of simulated air concentration measurements and gamma dose rate measurements have been created as described in Section 2.1. The same meteorological data and dispersion model have been used for data creation and for the source term estimation. Thus, the study demonstrates the uncertainties of the estimated source term arising only from the information loss due to the limited measurement capabilities. Due to the idealized nature of our study, our results apply to a situation, where model errors are negligible. In reality, meteorological uncertainties and model errors will further increase the uncertainty of the estimated source term.

Section 2 describes the data and methodology; Section 2.1 describes the synthetic measurement data set, Sections 2.2 and 2.3 describe the meteorological data and the dispersion model used, while Sections 2.4–2.7 describe the methodology. Next, the results are presented and discussed in Section 3. Finally, Section 4 presents a summary and the conclusions of the study.

## 2. Materials and Methods

### 2.1. Artificial Loviisa Release Case

For the artificial release from the Loviisa nuclear power plant in south Finland, the selected source term describes a core melt event without functioning mitigation systems. The initial event is a total loss of all power systems without battery back-up. The filtered containment venting system is assumed disconnected, and instead comprises an exhaust pathway from the reactor containment. It is postulated that the exhaust pathway was open at the time of melt-through of the reactor vessel. The released activity was corrected for decay and ingrowth for the time period between the emergency shutdown of the nuclear

reactor (SCRAM) and the time of the release starting three hours later. It is assumed that there was no significant heat release associated with the accident, and therefore all material is released from a fixed height of 27 m above ground.

The time evolution is given in one-hour time steps starting at the onset of the accident (time of the SCRAM) and the following 12 h, intended to represent the first part of the release to undergo subsequent detection by the gamma monitoring stations and capture by the air filter stations. The source term was developed for the research project SOurce CHAracterizatiOn accounting for meTeorologIcal unCertainties (SOCHAOTIC), for further details, see [11].

Figure 1 shows the gamma dose field at the end of the simulation, 63 h after the release starts, as well as the locations of gamma dose rate stations and filter stations. The source term is given in Section 3.

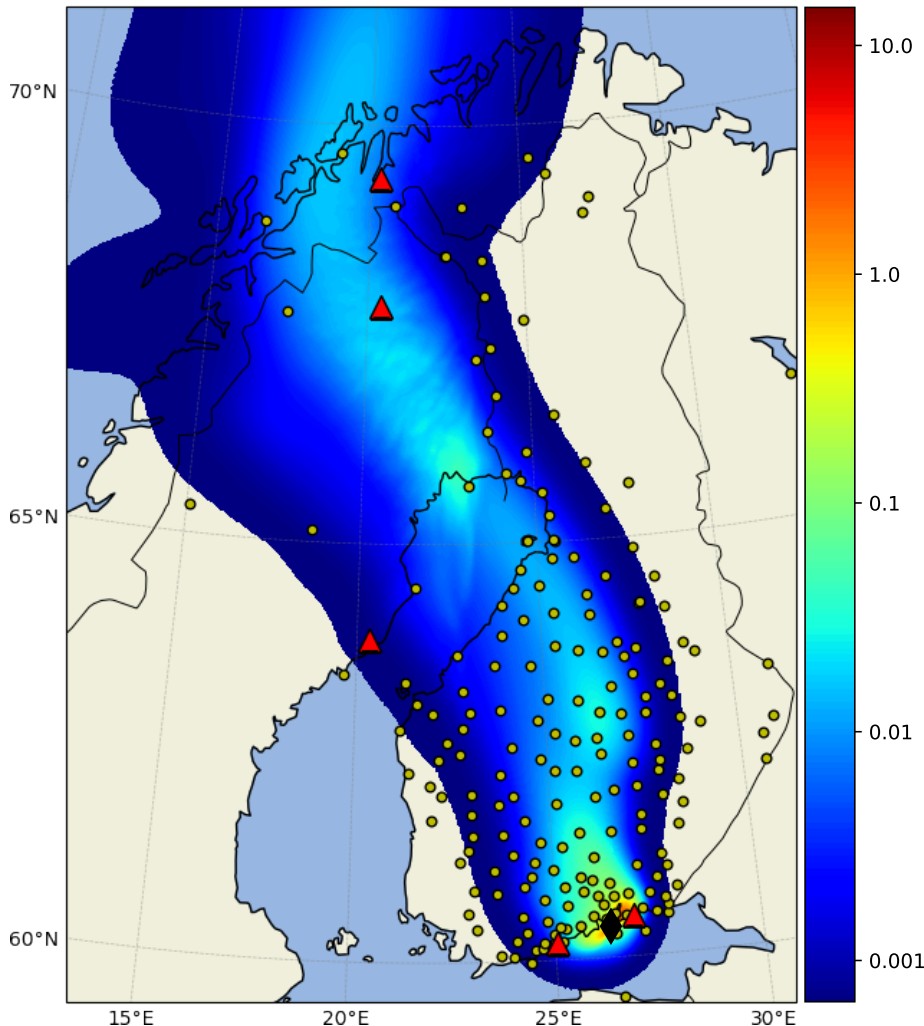

**Figure 1.** Total gamma dose in units of mSv at 63 h after the release start. Areas only influenced by background radiation are left uncolored. The black diamond shows the release location, the yellow circles show the locations of the gamma stations, and the red triangles show the locations of the filter stations.

### 2.1.1. Simulated Gamma and Filter Station Measurements

The total dose rate at the gamma monitoring stations is the sum of the contributions from cloud and ground since the stations are not shielded from activity deposited on the ground. Over time, the contamination of the station itself will also contribute to the measurements.

A set of 11 nuclides was selected to represent the most important nuclides for human doses: Kr-88*, Xe-133*, Xe-135*, Xe-135m*, Cs-134#, Cs-137, I-131#, I-132*#, I-133#, I-135# and Te-132. The list consists of the expected top five for the gamma monitoring stations (denoted by ∗), and top five for the air filter stations (denoted by #), expected to represent more than 90% of the dose rate contribution in the first 12 h of the postulated event. Moreover, two nuclides from the top ten list, Cs-137 and Te-132, were included since they represent key nuclides as seen from historical releases. For further details, see [11].

The artificial scenario consisting of simulated filter station and gamma station measurements was derived by predicting the atmospheric dispersion of radionuclides from a 9-hour release at the Loviisa nuclear power plant starting at 08:00 UTC on 22 September 2021. The DERMA atmospheric dispersion model was applied to the release scenario described above and using Harmonie data, cf. Sections 2.2 and 2.3, thereby providing average concentration values at existing filter stations, and gamma dose rates at gamma stations by using the ARGOS gamma dose model [12,13]. The filter concentration values are computed as 24 h averages from 08:00 UTC to 08:00 UTC the next day. Further, the filter measurement stations are assumed to have a detection limit of 0.1 mBq m$^{-3}$. For the gamma dose rates, we have assumed a background radiation of 0.1 μSv h$^{-1}$, which has been added to all modelled dose rates.

## 2.2. Meteorological Data

The simulations have been carried out using meteorological data derived by the non-hydrostatic convection-permitting limited-area numerical weather prediction model Harmonie [14]. The horizontal grid resolution is approximately 2.5 km, and the vertical dimension is resolved by 65 levels with a terrain-influenced hybrid coordinate. The lowest model level is about 12 m above ground, and the highest at approximately 10 hPa. The model is configured with three-hourly data assimilation cycling. For the Loviisa case, the model simulation starts on 22 September 2021, at 00:00 UTC and runs until 24 September 2021, at 23:00 UTC.

## 2.3. Dispersion Modelling

The atmospheric dispersion is modelled by using the Danish Emergency Response Model of the Atmosphere (DERMA) [15,16]. DERMA is used operationally for a number of Danish emergency preparedness purposes [17–21] including nuclear [13]. The three-dimensional model is of Lagrangian type making use of a hybrid stochastic particle-puff diffusion description [15,16]. The model uses aerosol size dependent dry and wet deposition parameterizations as described by [22].

DERMA is interfaced with the nuclear decision-support system ARGOS (Accident Reporting and Guidance Operational System) [12,13], where the integration is accomplished through automatic online exchange of data between ARGOS and the DMI High Performance Computing (HPC) facility. The dose calculation modules are incorporated in ARGOS.

## 2.4. Problem Description

The temporal release profiles of the radionuclides considered are estimated by using observations of both air concentration and gamma dose rate combined with a series of forward runs by the dispersion model DERMA. We assume an overall start time $t_0$ and end time $t_n$ of the release. We then separate the total release period into $n$ time bins of duration $\Delta t_{\text{bin}}$ and for each of these assume a unit release of each of the included radionuclides. The releases are assumed to be point releases at ground level. As described in Section 2.1.1, we assume that only a selection of all released radionuclides contributes significantly to the gamma dose rates, while other radionuclides will be ignored. Let $C^{\text{o}}_{ik}$ be the $k$'th observed average concentration of the $i$'th radionuclide, measured over a specified time period at a specified filter station. Similarly, let $\Gamma^{\text{o}}_{\kappa}$ be the $\kappa$'th observed gamma dose rate measured at a specified time and gamma station.

The atmospheric dispersion model DERMA is run forward in time for each of the unit releases, and for the $j$'th release of the $i$'th radionuclide the average activity concentrations $\bar{c}_{ijk}$ are calculated, where the $k$-index corresponds to the location and time of an existing filter measurement. Further, instantaneous activity concentrations $c_{ij\kappa}$ and deposition values $d_{ij\kappa}$ are calculated, where the $\kappa$-index corresponds to the location and time of an existing gamma dose rate observation. By using the gamma dose model as described in Section 2.3, the contributions to the gamma dose rates $\gamma_{ij\kappa} = \gamma_{ij\kappa}(c_{ij\kappa}, d_{ij\kappa})$ are calculated. For a given set of non-negative coefficients, $\lambda_{ij}$, the predicted average concentrations and gamma dose rates corresponding to existing measurements are calculated:

$$C_{ik}^{\mathrm{m}} = \sum_j \lambda_{ij} \bar{c}_{ijk}$$

$$\Gamma_{\kappa}^{\mathrm{m}} = \sum_i \sum_j \lambda_{ij} \gamma_{ij\kappa}. \tag{1}$$

*2.5. Bayesian Inversion and Sampling Method*

Given a set of observations, $(\mathbf{C}^{\mathrm{o}}, \mathbf{\Gamma}^{\mathrm{o}})$, the coefficients, $\boldsymbol{\lambda}$ can be determined by applying Bayes' theorem:

$$P(\boldsymbol{\lambda}, \boldsymbol{\theta} | \mathbf{C}^{\mathrm{o}}, \mathbf{\Gamma}^{\mathrm{o}}, I) = \frac{P(\boldsymbol{\lambda}, \boldsymbol{\theta} | I) \, P(\mathbf{C}^{\mathrm{o}}, \mathbf{\Gamma}^{\mathrm{o}} | \boldsymbol{\lambda}, \boldsymbol{\theta}, I)}{P(\mathbf{C}^{\mathrm{o}}, \mathbf{\Gamma}^{\mathrm{o}} | I)}, \tag{2}$$

where $\boldsymbol{\theta}$ denotes any so-called nuisance parameters, i.e., unknown parameters, which are not of direct interest. One way to account for these is to treat them just like the parameters of interest and consider $P(\boldsymbol{\lambda}, \boldsymbol{\theta} | \mathbf{C}^{\mathrm{o}}, \mathbf{\Gamma}^{\mathrm{o}}, I)$, which is the posterior probability distribution for the combined set of parameters $(\boldsymbol{\lambda}, \boldsymbol{\theta})$. $P(\boldsymbol{\lambda}, \boldsymbol{\theta} | I)$ is then the prior probability distribution for $(\boldsymbol{\lambda}, \boldsymbol{\theta})$, $P(\mathbf{C}^{\mathrm{o}}, \mathbf{\Gamma}^{\mathrm{o}} | \boldsymbol{\lambda}, \boldsymbol{\theta}, I)$ is the likelihood, and $P(\mathbf{C}^{\mathrm{o}}, \mathbf{\Gamma}^{\mathrm{o}} | I)$ is the evidence; a statistical constant independent of $(\boldsymbol{\lambda}, \boldsymbol{\theta})$. $I$ is any background information that may be available, e.g., amount of material present in the core at the time of the accident.

To evaluate Equation (2), the quantities $P(\boldsymbol{\lambda}, \boldsymbol{\theta} | I)$ and $P(\mathbf{C}^{\mathrm{o}}, \mathbf{\Gamma}^{\mathrm{o}} | \boldsymbol{\lambda}, \boldsymbol{\theta}, I)$ must be estimated for a selection of realizations of $(\boldsymbol{\lambda}, \boldsymbol{\theta})$, and the resulting posterior probability distribution $P(\boldsymbol{\lambda}, \boldsymbol{\theta} | \mathbf{C}^{\mathrm{o}}, \mathbf{\Gamma}^{\mathrm{o}}, I)$ can then be estimated by normalizing the distribution. The posterior probability distribution for $\boldsymbol{\lambda}$ can then be determined by marginalizing:

$$P(\boldsymbol{\lambda} | \mathbf{C}^{\mathrm{o}}, \mathbf{\Gamma}^{\mathrm{o}}, I) = \int_{\boldsymbol{\theta}} P(\boldsymbol{\lambda}, \boldsymbol{\theta} | \mathbf{C}^{\mathrm{o}}, \mathbf{\Gamma}^{\mathrm{o}}, I) \, \mathrm{d}\boldsymbol{\theta}. \tag{3}$$

To get a good estimate of the probability distribution, the relevant parts of the parameter space must be sampled. One option is to use random-walk based Markov Chain Monte Carlo (MCMC) methods, such as Metropolis-Hastings or Gibbs [23,24]. However, these methods generally require a large number of iterations, because the random-walk based model proposals do not sample the parameter space of the posterior probability distribution in the most efficient way. Further, parameters such as the step size of the random-walk typically need to be tuned to the specific case. Instead, we use the Hamiltonian Monte Carlo (HMC) method No U-Turn Sampling (NUTS) [25], implemented in the Python library PyMC3 [26]. HMC methods generally have an advantage over random-walk based MCMC methods, because the model proposals are not generated by a random-walk but instead based on estimated gradients of the posterior distribution. Thus, much fewer iterations are typically needed to sufficiently sample the probability distribution. However, the efficiency of HMC algorithms strongly depends on the step size parameter. The NUTS algorithm uses adaptive step sizing such that the step size does not need to be set by the user. Further, as the name suggests, the algorithm is constructed such that trajectories in the parameter space avoid making "U-turns", i.e., retracing their own steps. Thus, it should produce more independent samples in fewer iterations. When the aim is to use Bayesian inverse modelling operationally, the NUTS algorithm is ideal, since very little parameter tuning is necessary [25]. In addition, when using the PyMC3 implementation [26], Gelman-Rubin convergence diagnostics [27] are automatically calculated, when sampling with two

or more chains. This makes it easy to control that the sampler has converged. For further details on the NUTS algorithm, see [25].

### 2.6. Prior Probability Distributions

Defining useful prior probability distributions for the release rates is challenging, since the magnitude of the release is unknown. To allow for variation over several orders of magnitude while ensuring non-negative values, we use log-normal prior distributions. Assuming a normal distributed variable $x \sim \mathcal{N}(\mu, \sigma)$, then the variable $z = e^x \sim \text{Lognormal}(\mu, \sigma)$ is log-normal distributed with parameters $\mu$ and $\sigma$. Thus, these denote the mean and standard deviation of $x$ and not of the log-normal distributed variable $z$. The prior probability distribution for the coefficients $\lambda_{ij}$ can be written as:

$$P(\lambda_{ij}|I) = \text{Lognormal}(\mu_i, \sigma_i), \tag{4}$$

where $\mu_i$ and $\sigma_i$ are parameters to be determined for the specific radionuclide. Given that total amount of the $i$'th radionuclide in the core inventory is $S_i$ in units of Bq, the upper limit for $\lambda_{ij}$ is $S_i/\Delta t_{\text{bin}}$, where $\Delta t_{\text{bin}}$ is the duration in seconds of each assumed unit release. To allow for release rates approaching the upper limit with reasonable probability, we set $\mu_i + 2\sigma_i = \log(S_i/\Delta t_{\text{bin}})$, where $\log()$ denotes the natural logarithm. The lower limit must be small compared the "typical" release rate, $\mu_i$. Since the typical release rate is unknown, we assume $\mu_i = \log(fS_i/\Delta t_{\text{bin}})$, where $f$ is some (small) fraction. Assuming a sufficiently low value for $f$ will result in a conservative prior distribution, which allows for a broader range than necessary. In this study, we use $f = 10^{-3}$, which means that $\mu_i \pm 2\sigma_i$ includes six orders of magnitude for each release rate. Thus, the mean and standard deviations for the prior probability distributions are given as:

$$\mu_i = \log(10^{-3} S_i/\Delta t_{\text{bin}}) \quad \text{and} \quad \sigma_i = \frac{1}{2}\log(10^3). \tag{5}$$

Further, we can use information about the core inventory to reduce the parameter space by imposing correlations between release rates of certain radionuclides, inspired by the method by Saunier et al. [9,10]. For example, two different isotopes of the same element will largely behave similarly during a release. Thus, if the half-lives of two such isotopes are long compared to the duration of the release and if there is no significant ingrowth from other processes, the ratio of the release rates between two isotopes can be assumed constant and equal to the ratio of the amounts in the core inventory. For example, $^{134}$Cs and $^{137}$Cs have half-lives of approximately 2 and 30 years, respectively, and thus, the ratio of their activity concentrations in the core inventory can be considered constant during the release. Accordingly, based on the amounts of the two isotopes in the core, we can assume the ratio of their release rates to be constant.

For other isotope pairs, it is necessary to take into account the difference in half-lives in order to set realistic constraints on the release rates. In this case, knowing the amount of the two isotopes at the time of SCRAM gives one limit for the isotopic ratios, while estimating the activity concentration $n$ hours later will provide an estimate of the other limit, assuming no significant ingrowth. An example is the isotope pair $^{131}$I and $^{133}$I, which has half-lives of approximately 8 days and 20.8 h, respectively. Let the release rates of these isotopes be $q_{^{131}\text{I}}$ and $q_{^{133}\text{I}}$, respectively. Based on their activity concentrations in the core at the time of the accident, we have $\frac{q_{^{133}\text{I}}}{q_{^{131}\text{I}}} < 2.1$. Assuming that the duration of the main release is less than 24 h, we can determine the other limit. Due to radioactive decay during these 24 h, the amount of $^{133}$I is decreased by a factor of 0.45, while we assume that the amount of $^{131}$I is unchanged due to its relatively long half-life. Thus, a lower limit can be determined $\frac{q_{^{133}\text{I}}}{q_{^{131}\text{I}}} > 0.9$. Following this approach, we determine the following constraints:

$$\frac{q_{^{134}\text{Cs}}}{q_{^{137}\text{Cs}}} = 1.4, \quad 0.001 < \frac{q_{^{132}\text{I}}}{q_{^{131}\text{I}}} < 1.5, \quad 0.9 < \frac{q_{^{133}\text{I}}}{q_{^{131}\text{I}}} < 2.1 \quad \text{and} \quad 0.15 < \frac{q_{^{135}\text{I}}}{q_{^{131}\text{I}}} < 2.0. \tag{6}$$

For $^{134}$Cs and $^{137}$Cs, this effectively means that only one release rate needs to be determined instead of two, and that the combined set of measurements of the two isotopes can be used. For the other isotope pairs, the constraints allow us to define log-normal distributions with upper and lower bounds, which depend on the release rate of one of the other nuclides. Let $\lambda_{\mathrm{m}j}$ and $\lambda_{\mathrm{n}j}$ be the coefficients for two release rates, which are related by the flexible constraints $r_{\mathrm{lower}} < \lambda_{\mathrm{n}j}/\lambda_{\mathrm{m}j} < r_{\mathrm{upper}}$. Then, the prior probability distribution for $\lambda_{\mathrm{m}j}$ will be defined as in Equation (4), while the prior probability distribution for $\lambda_{\mathrm{n}j}$ can be written as:

$$P(\lambda_{\mathrm{n}j}|I, \text{constraints}) \propto \begin{cases} P(\lambda_{\mathrm{n}j}|I) & r_{\mathrm{lower}} < \frac{\lambda_{\mathrm{n}j}}{\lambda_{\mathrm{m}j}} < r_{\mathrm{upper}} \\ 0 & \text{otherwise} \end{cases}. \tag{7}$$

It might be possible to impose further constraints, i.e., across the type of element, such that the release rates of the iodine isotopes can also be related to the release rates of the caesium isotopes, Te-132 and the noble gasses. However, the underlying assumptions in this case are less trivial.

### 2.7. Likelihood and Uncertainty Quantification

The likelihood is the probability of observing the set of measurements $(\mathbf{C}^{\mathrm{o}}, \boldsymbol{\Gamma}^{\mathrm{o}})$, given a proposed source term, $\boldsymbol{\lambda}$. The likelihood is evaluated by assuming a probability distribution for the residuals $C_{ik}^{\mathrm{o}} - C_{ik}^{\mathrm{m}}(\lambda_{ij})$ and $\Gamma_{\kappa}^{\mathrm{o}} - \Gamma_{\kappa}^{\mathrm{m}}(\lambda_{ij})$. In this study, we use a log-normal likelihood, which is less sensitive to outliers than the Gaussian distribution and automatically gives a higher weight to measurements/predictions of low values. This makes it useful when dealing with measurement values over several orders of magnitude [6]. One practical challenge when dealing with log-normal distributions is that only positive values are mathematically allowed, while the physical quantity may in principle be zero. For the gamma dose rates, this is not an issue, since we add background radiation to the modelled measurements, thereby ensuring that values are always positive. For the air concentration measurements, on the other hand, modelled predictions may be zero, while the measured predictions may be below the detection limit. Assume that for a given measurement, $C_{ik}^{\mathrm{o}}$, the detection limit is $\epsilon_{ik}$. To avoid zero-values, we use these altered observations and model predictions $\widetilde{C_{ik}^{\mathrm{o}}} = \max(\epsilon_{ik}, C_{ik}^{\mathrm{o}})$ and $\widetilde{C_{ik}^{\mathrm{m}}} = \max(\epsilon_{ik}, C_{ik}^{\mathrm{m}})$. These altered forms have the additional benefit that they provide a theoretically sound way of using non-detections, since these will only contribute to the likelihood, when the modelled concentration is above the detection limit. Thus, there is no risk of falsely interpreting a low value as a zero. The likelihood is given as:

$$P\left(\widetilde{\mathbf{C}^{\mathrm{o}}}, \boldsymbol{\Gamma}^{\mathrm{o}}|\boldsymbol{\lambda}, I\right) = \prod_k \prod_i \text{Lognormal}\left(\widetilde{C_{ik}^{\mathrm{m}}}, \sigma_{\mathrm{f}}\right) \prod_{\kappa} \text{Lognormal}\left(\Gamma_{\kappa}^{\mathrm{m}}, \sigma_{\mathrm{g}}\right), \tag{8}$$

where $C_{ik}^{\mathrm{m}}$ and $\Gamma_{\kappa}^{\mathrm{m}}$ are as defined in Equation (1). $\sigma_{\mathrm{f}}$ and $\sigma_{\mathrm{g}}$ are related to the uncertainty of the measurements as well as the unknown model errors. In this study, both are negligible as discussed in Section 1. However, in order to make the method as general as possible, the uncertainty parameters are treated as nuisance parameters, i.e., they are kept as free parameters and sampled by the Monte Carlo algorithm. In practice, a wide uniform distribution has been used as prior distribution for the nuisance parameters $\sigma_{\mathrm{f}}, \sigma_{\mathrm{g}} \sim U(0, 10)$, which allows for a broad range of shapes of log-normal distributions.

## 3. Results and Discussions

As described in Section 2.5, the results are obtained by using the NUTS algorithm [25], which is implemented in the PyMC3 python library [26]. The algorithm is constructed in such a way that almost no parameter tuning is necessary. To ensure convergence, the target acceptance rate was increased from the default 0.8 to 0.99. Aside from this, everything was kept at PyMC3's default values; two simultaneously running chains, each with 1000 tuning steps and 1000 draws from the target distribution. This provides a total of 2000 realizations

of the posterior probability distribution. For further details on the NUTS parameters, see [25,26].

In our analysis, we include 10 of the 11 radionuclides described in Section 2.1.1, excluding Xe-135m based on the rationale that its short half-life of approximately 15 min makes it unimportant on longer temporal, and thus also spatial scales. This means that there is not enough information in the measurement data to sufficiently constrain the release rate of Xe-135m. The other three noble gasses are included, although there are no filter measurements to help constrain their release rates. However, as long as their half-lives are sufficiently different, we expect the gamma dose rate patterns to differ enough to be able to distinguish between their contributions. The prior probability distributions for the release rates of Kr-88, Xe-133, Xe-135, Cs-137, I-131 and Te-132 were defined as log-normal distributions, Equation (4) with mean and standard deviations given by Equation (5). The release rate for Cs-134 was defined as a deterministic variable, equal to the release rate for Cs-137 multiplied by the fixed ratio 1.4. Finally, the prior distributions for the release rates of I-132, I-133 and I-135 were defined as bound log-normal distributions Equation (7), where the bounds are given by the flexible constraints, Equation (6).

We assume that the time of the emergency shutdown of the nuclear reactor (SCRAM), 22 September, 05:00 UTC, is known. We therefore consider this as the first possible time of release. We then consider the release during the following 24 h by assuming twelve 2-h constant releases, i.e., $\Delta t_{bin} = 7200$ s. The source term estimation is based on the simulated measurements described in Section 2.1.1, but only measurements until 23 September, 08:00 UTC are used for the source term estimation, leaving the remaining measurements for validation of model predictions based on the estimated source term. Thus, for all particles, only two 24-h filter measurements from each of the five filter stations are available, i.e., ten filter measurements per particle. However, first, all measurements without any information are discarded; if a given measurement is not influenced by any of the time-binned unit releases, it is removed from the data set. After this automatic removal of data, only one filter measurement per particle from each of the two filter stations in southern Finland are left. Thus, even when using the additional constraints described in Section 2.6, the amount of filter measurement data is very limited.

The gamma dose rates, on the other hand, are measured every hour at 214 different locations, see Figure 1. Thus, from 22 September, 05:00 UTC to 23 September, 08:00 UTC, a total of 5778 measurements. After the automatic removal of data without information, 1918 measurements are left.

Given the high dimensionality of the parameter space, it is not possible to visualize all elements of the actual posterior distribution. Instead the individual release rates are shown in Figure 2. The plots show the median release rates as well as the 10th and 90th percentiles based on marginal distributions for each 2-h release period. Further, Figure 3 shows histograms of the marginal distributions of time integrated releases for all radionuclides. The only release rate, which is well determined for most time bins is that of Xe-133. This makes sense, since it is the only relatively long-lived noble gas; the half-life is approximately five days, while Xe-135 and Kr-88 have half-lives of roughly nine and three hours, respectively. Further, since the noble gasses do not deposit, the gamma dose rate pattern of Xe-133 will also be easy to distinguish from those of the long-lived particles. For the particles, the estimated release rates clearly indicate the effects of the constraints in Equation (6); the release rates of the four iodine isotopes, which are all "tied together", are better estimated than those of both the caesium isotopes and of Te-132. Since the release rates of the two caesium isotopes are forced to differ only by a factor, we also expect these to be better estimated than the release rate of Te-132. While it is not easy to see that this is the case, it is clear from Figure 3 that the released amounts of the two caesium isotopes are better estimated than Te-132.

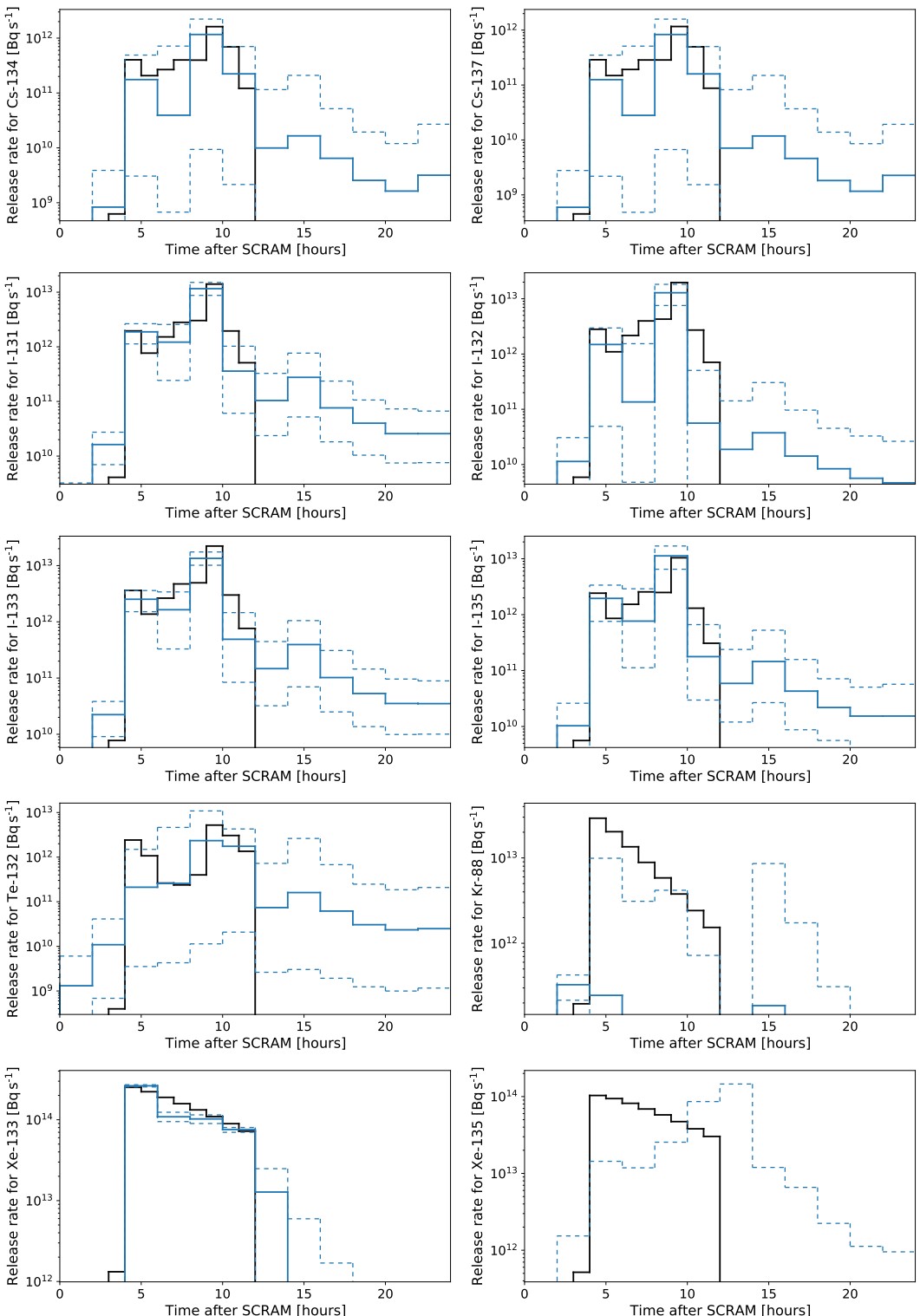

**Figure 2.** Release rates for each radionuclide in each 2-h time bin. The solid blue lines show the medians of the marginal distributions, while the dashed blue lines show the 10th and 90th percentiles. For comparison, the solid black lines show the true release profile. To focus on the release rates of high magnitude, we have set the minimum value on the *y*-axis to 10% of the lowest true release rate. Thus, predicted release rates below this limit are not shown in the plot, e.g., the predicted release rate of Xe-135 only shows the 90th percentile, while both the 10th percentile and the median are below the axis limit.

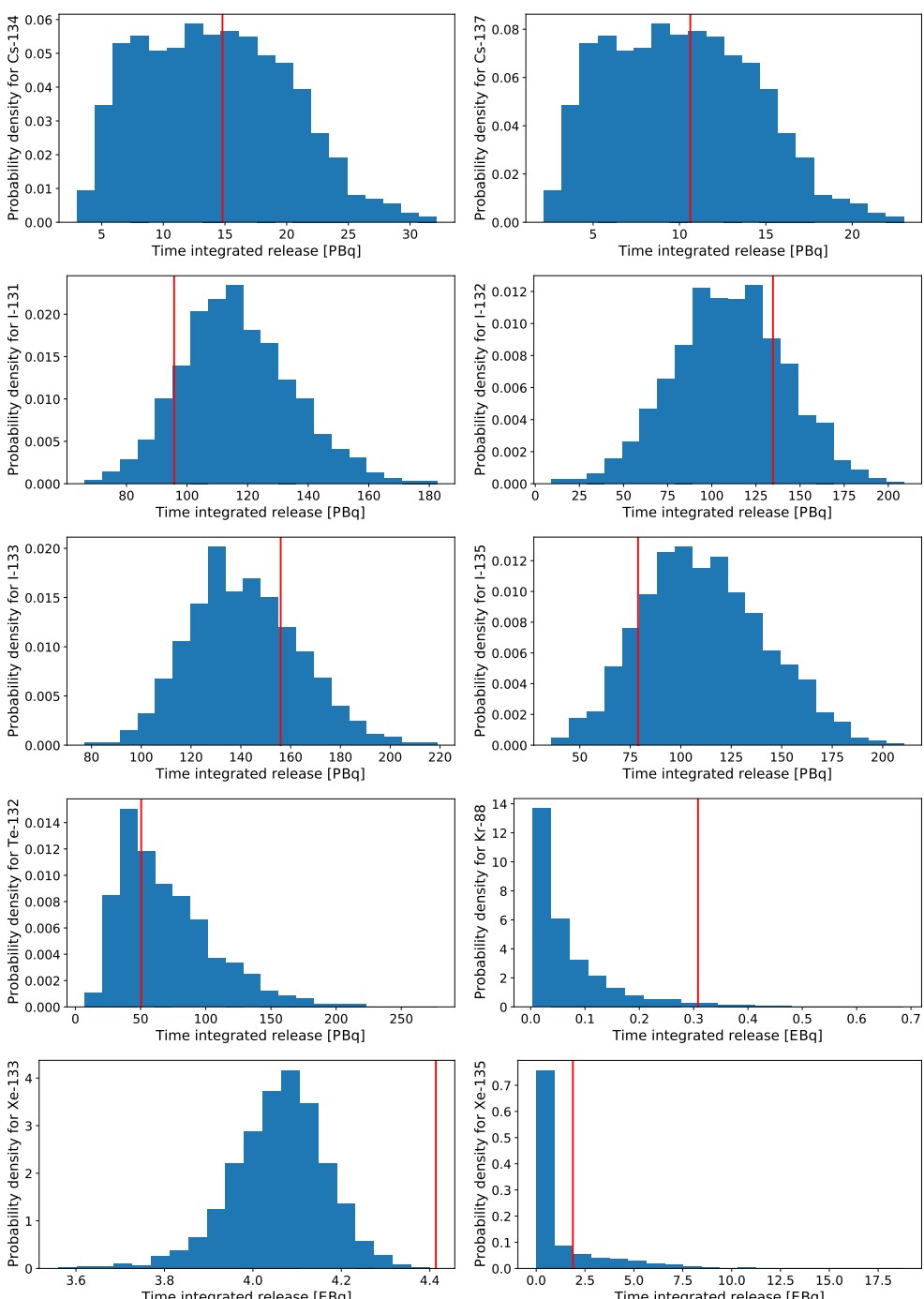

**Figure 3.** Probability density for each radionuclide as a function of time-integrated release. The vertical red lines show the actual released amounts.

The histograms in Figure 3 show that for some radionuclides, the amounts are quite well constrained, e.g., the release of I-131, which varies from roughly 70 PBq to 180 PBq, and Xe-133, which varies from roughly 3.6 EBq to 4.4 EBq. The latter, however, only barely include the true released amount in the probability distribution. For the remaining radionuclides, the released amounts are not very accurately estimated, especially not for Kr-88 and Xe-135. Given the limited amount of measurement data, this result is not surprising. Further, it is important to note that the log-normal prior distribution ensures release rates of positive values. Hence, the estimated release will necessarily have the same duration as the considered release period, 24 h in this case. However, we see from Figure 2 that most release rates drop significantly in magnitude after 12 h from SCRAM.

From Figures 2 and 3, it may seem that the source term is not sufficiently constrained by the data. Clearly, release rates for some nuclides are poorly estimated, e.g., Kr-88 and Xe-135, and it may therefore be tempting to exclude these from the source term. However, we found that when excluding these, the estimated release rates of the remaining nuclides are less accurate. Thus, it seems that the release of some of the other nuclides compensate for their lacking contribution. On the other hand, it is important to note that including Kr-88 and Xe-135 in the source term does not seem to compromise the release rates of the remaining nuclides. Thus, when it is not known a priori which nuclides constitute the best possible source term, the safer choice seems to be to include more nuclides than necessary. Further, the marginal distributions are obtained by integrating over the remaining parameters of the multi-nuclide source term, and therefore all correlations between parameters are ignored. As demonstrated below, though the marginal distributions of individual releases might be uncertain, the gamma dose rate patterns of different realizations of the multi-nuclide source term vary significantly less.

Figure 4 shows predicted air concentrations and gamma dose rates as function of observations. The upper plots show filter measurements, and the lower plots show gamma dose rates. The left plots show measurements before 23 September, 08:00 UTC, i.e., the measurements that are used for the source term estimation. The right plots show measurements after 23 September, 08:00 UTC and therefore show a prediction of future values based on the estimated source term. The percentiles are estimated by first calculating the concentrations and gamma dose rates from all source terms in the posterior distribution and then finding the percentiles in the calculated values. The plots with the gamma dose rates show a randomly selected subset of 300 observations, since more data in the plot makes it impossible to distinguish the different data points. The figure shows that the average activity concentrations at the filter stations are generally estimated to match the observations within the uncertainties, although some allow for a wide variation. On the other hand, the predicted gamma dose rates fit very well with the observed even for the predicted values. Considering the fact that a total of 1918 gamma measurements and only 2 filter measurements for each nuclide are used for the inversion, it is not surprising that the gamma dose rates are more accurately estimated.

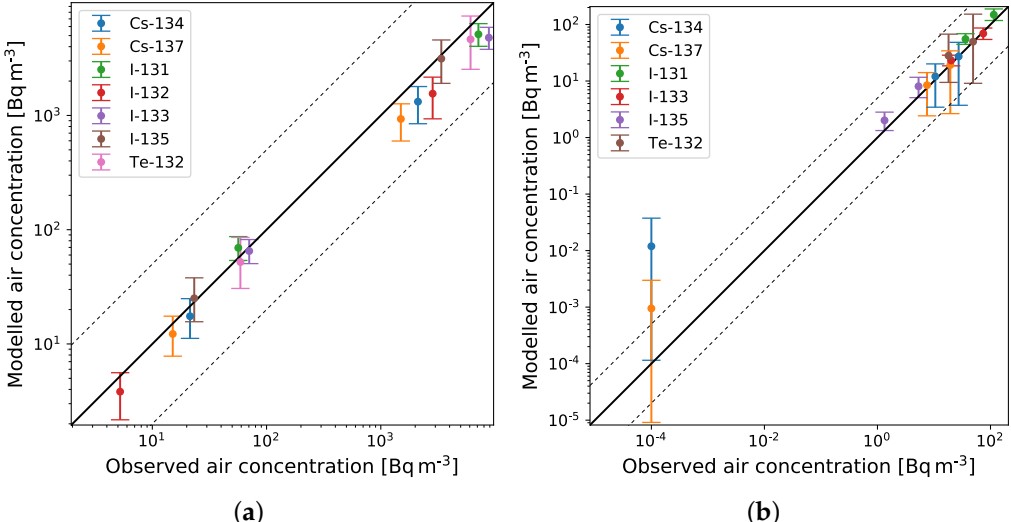

**Figure 4.** *Cont.*

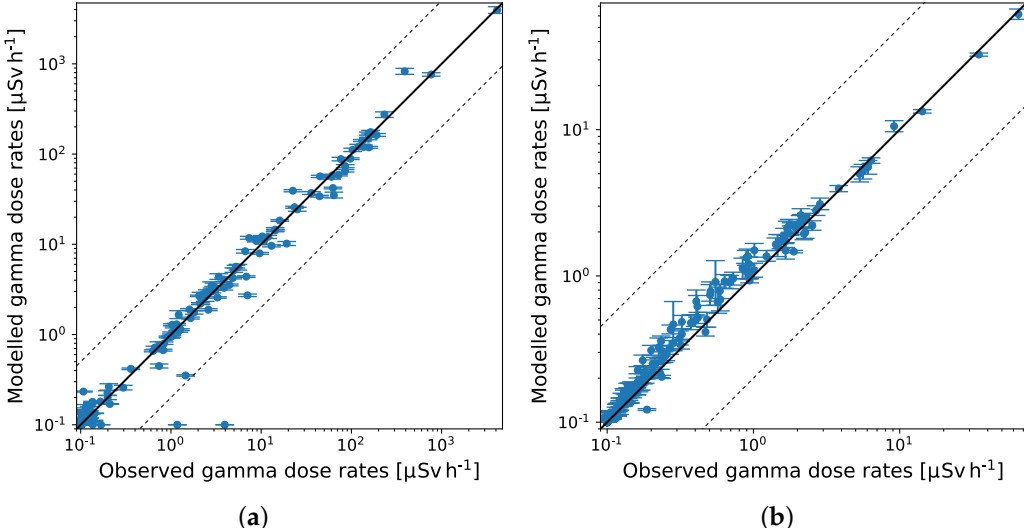

(**a**) (**b**)

**Figure 4.** Model predictions with uncertainties (median and 10th and 90th percentile) on the *y*-axis, and observations on the *x*-axis. The solid black lines indicate a perfect correlation, while the dashed black lines indicate a factor of 5 between model and observation. (**a**) shows the filter measurements until 23 September, 08:00 UTC, i.e., the measurements that are used for the source term estimation, whereas (**b**) shows the filter measurements after 23 September, 08:00 UTC, i.e., predicted future air concentrations. (**c**) similarly shows the gamma dose rates until 23 September, 08:00 UTC, and (**d**) shows the gamma dose rates after 23 September, 08:00 UTC.

Figure 5 shows the predicted gamma dose rates at the locations of six selected gamma stations, viz. the six stations that measured the highest values. The plots show that there is good agreement between modelled observed gamma dose rates and that even the time evolution is captured very well.

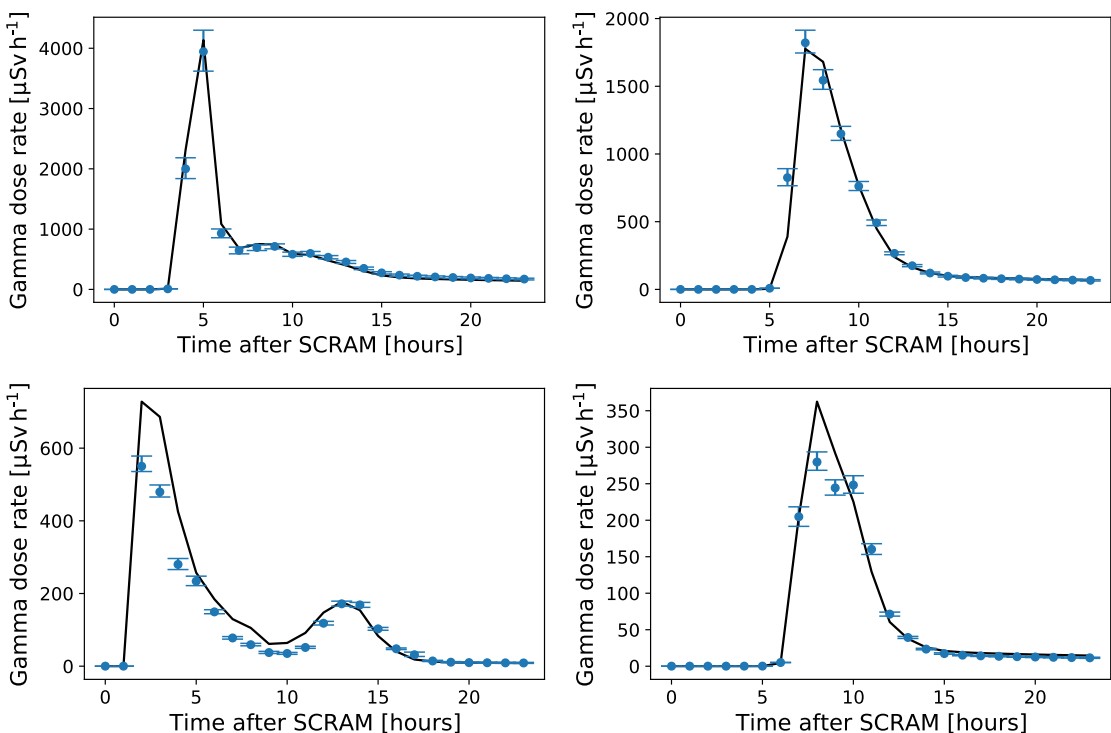

**Figure 5.** *Cont.*

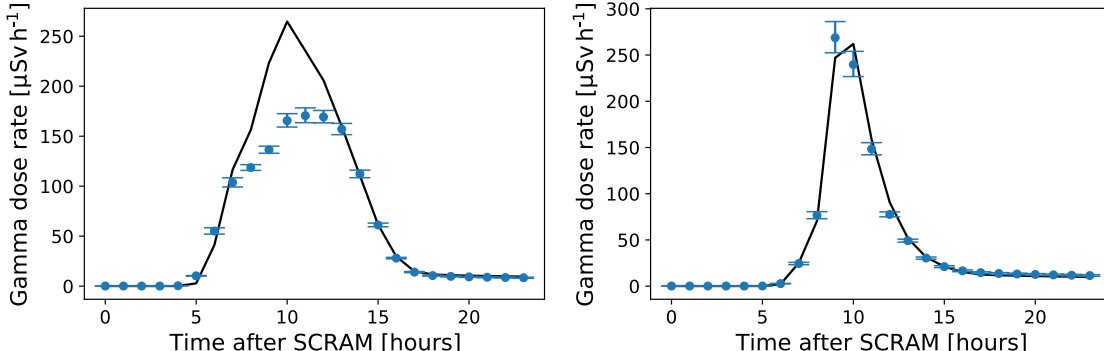

**Figure 5.** Gamma dose rates at locations of gamma stations during the first 24 h after the accident. Model predictions with uncertainties (median and 10th and 90th percentile) are shown by the blue dots and error bars, while the true gamma dose rates are shown by the black solid line. The selected gamma stations are all close to release locations, viz. the six stations that measured the highest values during the first 24 h.

Finally, Figure 6 shows the probability distributions of the two uncertainty parameters $\sigma_f$ and $\sigma_g$; both parameter distributions indicate relatively narrow log-normal distributions, which is expected given that model errors are negligible.

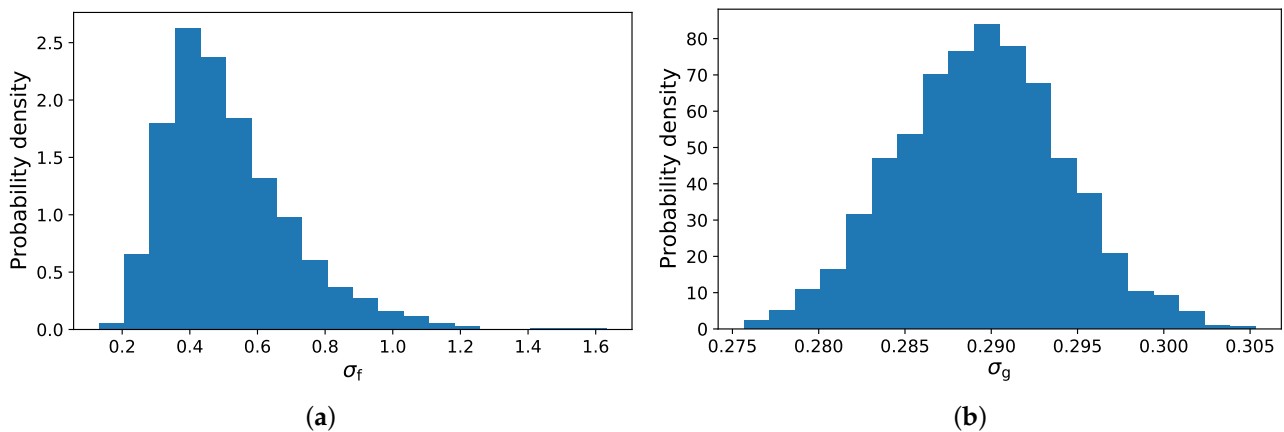

**Figure 6.** Marginal probability distributions of the uncertainty parameters, (**a**) $\sigma_f$ and (**b**) $\sigma_g$.

### 3.1. Including All Data

For comparison, we show the estimated source term when including all measurements. Figure 7 shows the release rates and probability densities of released amounts for three selected nuclides, Cs-134, I-131 and Xe-133. Interestingly, the release rates are all better defined than the previous result, i.e., the distributions are narrower. However, the release rate estimates are not necessarily more accurate. On the other hand, comparison with Figure 2 shows that the use of later measurements allows for a better estimate of the duration, as all release rates are very low after 16 h from the SCRAM.

As discussed previously, there are not many filter measurements available, and therefore the gamma dose rates are dominant; thus, the estimated source term is more likely to match the gamma dose rates than the filter measurements. This is apparent from Figure 8, which shows the modelled air concentrations and gamma dose rates as function of observations, similar to Figure 4. There is a very good agreement for gamma dose rates, while for air concentrations, the discrepancy is somewhat larger.

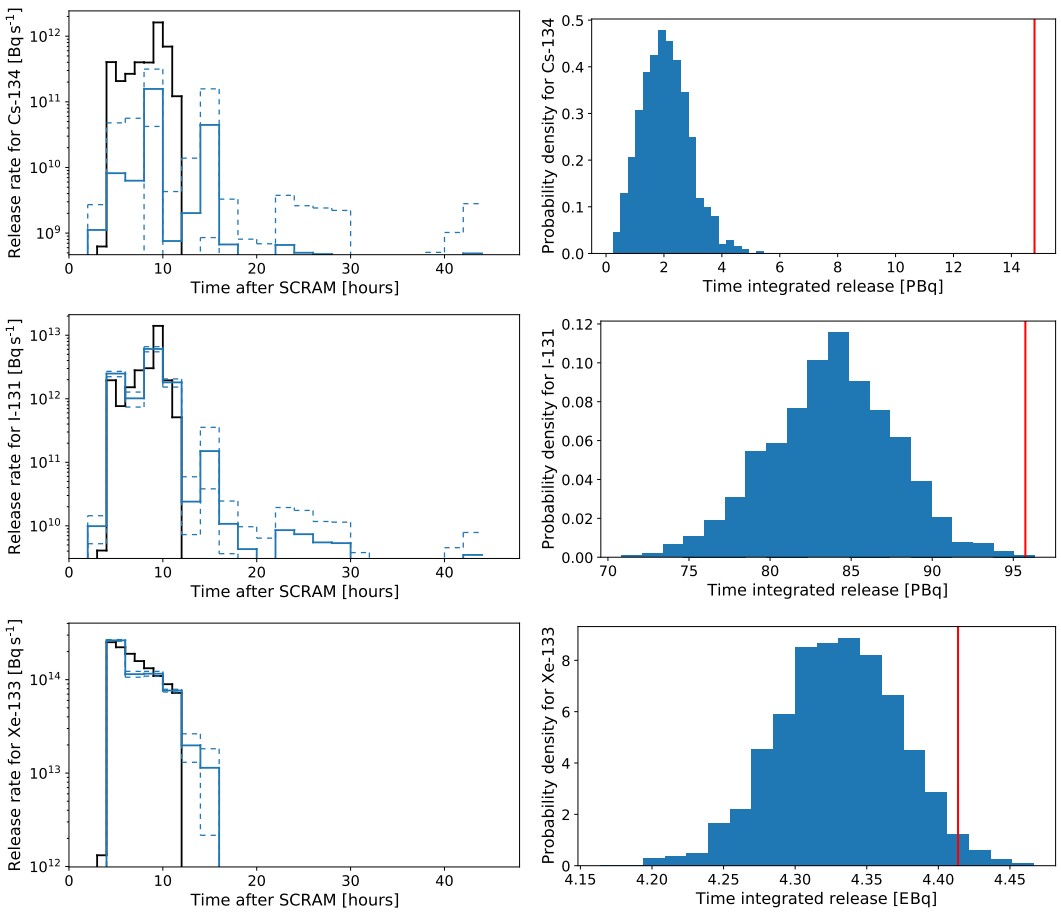

**Figure 7.** Release rates and probability densities for selected nuclides. For further description of the plots, see captions of Figures 2 and 3.

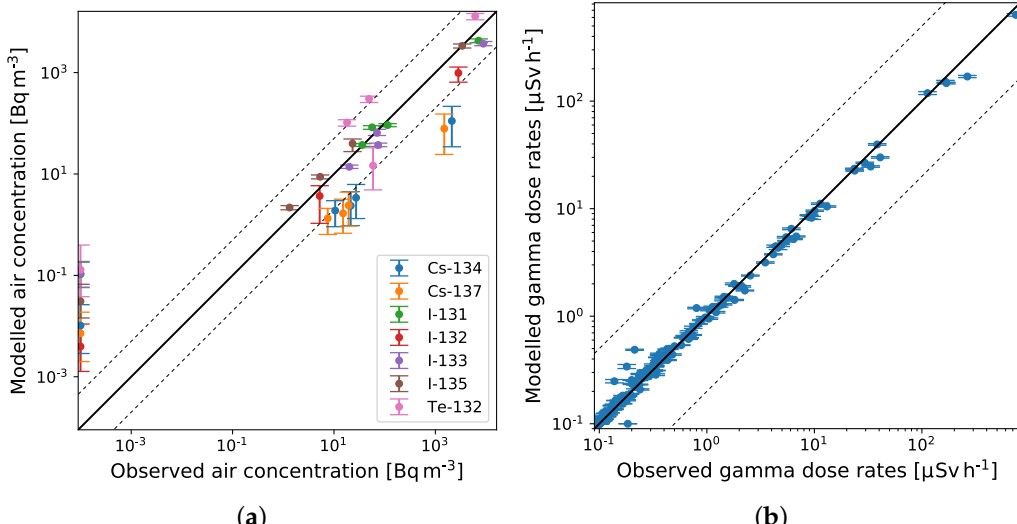

(**a**)　　　　　　　　　　　　　　(**b**)

**Figure 8.** Model predictions with uncertainties (median and 10th and 90th percentile) on the *y*-axis, and observations on the *x*-axis. The solid black lines indicate a perfect correlation, while the dashed black lines indicate a factor of 5 between model and observation. (**a**) shows the filter measurements, whereas (**b**) shows the gamma dose rates.

*3.2. Efficiency*

Regarding efficiency, we only have rough estimates of the computation time. However, we see that the time depend strongly on the amount of data included. The computation time for the first result, using data from only the first 24 h, was approximately half an hour. When including all data, the computation time was approximately 3.5 h. These estimates are the wall times of the runs of the NUTS algorithm, when running the algorithm in parallel on two CPUs on a standard modern laptop. In addition, some time is of course required for running the dispersion model and restructuring the data.

When operationalized, the code should be adapted to run on an HPC facility to further decrease computation time. In addition, the total set of gamma dose rate observations constitute 8953 measurements from a relatively dense network sampling at every hour. We suspect that there is a lot of redundant information in this data set, so instead using a subsample of this data set might be sufficient and would reduce computation time significantly.

## 4. Summary and Conclusions

We have developed a Bayesian inverse method for probabilistic source term estimation to be used for accidental nuclear releases to the atmosphere. The source term probability distribution is sampled using the Hamiltonian Monte Carlo algorithm NUTS, which is robust and needs only limited parameter tuning. In theory, this makes it directly applicable to other cases without making significant changes to the method.

The method is applied to a synthetic data set derived by running an atmospheric dispersion model for a realistic accident at a nuclear power plant. The data set consists of air concentration measurements at existing filter stations as well as gamma dose rates at gamma stations. We have shown that even with a limited set of air concentration measurements, realistic source term estimation is possible based on early observations of gamma dose rates. Further, the results indicate that additional constraints on the release rates based on information on the nuclear reactor core inventory can be used to improve the accuracy of the predictions. The estimated released amounts of most individual radionuclides are described by relatively wide probability distributions. However, the gamma dose rates predicted using the probabilistic source term correspond well with observations.

Of course, when applied to a real-world case, we expect that model errors will reduce the accuracy of the predictions to some extent. However, if the models used are unbiased, we anticipate that the predicted gamma dose rates will still be more accurately estimated than the release rates of the individual radionuclides. Further, to make the method as generally applicable as possible, we treat the uncertainty parameters as nuisance parameters. Hence, no assumptions about the magnitude of the uncertainties are made; the only assumption is that the residuals are log-normal distributed.

In conclusion, we have developed a method that performs well applied to the simulated release case, and the results indicate that even with limited measurement data available, it is possible to construct a probabilistic source term that provides accurate predictions of gamma dose rates and reasonable estimates of the released amounts of most of the radionuclides considered. Due to the few assumptions made and the robust theoretical foundation, we expect the method to generalize well. However, in order to fully examine the performance of the method, future application to real-world cases is necessary.

**Author Contributions:** Conceptualization, K.S.T. and J.H.S.; methodology, K.S.T.; software, K.S.T.; validation, K.S.T.; formal analysis, K.S.T.; investigation, K.S.T.; writing—original draft preparation, K.S.T.; writing—review and editing, K.S.T. and J.H.S.; visualization, K.S.T.; supervision J.H.S.; funding acquisition, K.S.T. and J.H.S. All authors have read and agreed to the published version of the manuscript.

**Funding:** This research was funded by the Innovation Fund Denmark grant number 0196-00017B, by the Danish Meteorological Institute, and by the Nordic Nuclear Safety Research (NKS) through the

SOCHAOTIC (SOurce CHAracterizatiOn accounting for meTeorologIcal unCertainties) project, grant number AFT/B(22)1.

**Institutional Review Board Statement:** Not applicable

**Informed Consent Statement:** Not applicable

**Conflicts of Interest:** The authors declare no conflict of interest.

## Abbreviations

The following abbreviations are used in this manuscript:

| | |
|---|---|
| DERMA | Danish Emergency Response Model of the Atmosphere |
| MCMC | Markov Chain Monte Carlo |
| HMC | Hamiltonian Monte Carlo |
| NUTS | No U-Turn Sampling |
| SOCHOTIC | SOurce CHAracterizatiOn accounting for meTeorologIcal unCertainties |

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
