# Peer review of "Bayesian Inverse Modelling for Probabilistic Multi-Nuclide Source Term Estimation Using Observations of Air Concentration and Gamma Dose Rate"

_atmosphere, doi:10.3390/atmos13111877_

Round 1

Reviewer 1 Report

Please see the attached file for suggestions and comments.

Author Response

Please find attached our responses to your comments.

Reviewer 2 Report

This is a carefully done study and the findings are of considerable interest. A few minor revisions are listed below. 

Figure.1 :

The caption of Figure 1  is poorly explained.  In line 95, the release time is not shown. So I don't understand what does the time Sep. 24 23:00 UTC mean.   In line 114, I know the release time.

-> For example,  this has been replaced by " Total gamma dose in units of mSv at 63 hours after the release starts."  

Figure.1 and Figure.5 :

 I see the purple triangles  as "red" triangles.  If so, the color of triangles should be changed.

Line 230 and Line 232:

The "q" is poorly defined. "0.45" is poorly explained. 

Figure.2 : 

In the release rate of Xe-135, I don't see the solid blue line. 

Figure. 3:

The red diamonds are hard to see. The red line is better.  

Figure.4 (a) and (b):

The dashed line indicates a factor of 5 between model and observation.

The fine solid line is correct?

Figure. 5 :

The maximum gamma dose rates and minimum one are shown in this figure. However, the difference is negligible. So, the median should be only indicated.   In line 360-363, difference the worst case and the best case is explained.

Author Response

(The authors gave the same response as above.)

Reviewer 3 Report

Though the paper has some points of interest, I think the general presentation of the research is very poor and needs several important modifications and clarification before it can be considered again for publication.

Manuscript title is long.

The abstract is weak. It is not specific and contains a lot of details about the case study description that are not useful for understanding the scientific issued addressed in this work.

Lack of motivation. The issue that the authors want to face is not adequately described, and it is too vague.

The references reported in the introduction are not representative of the state of the art related to flood mapping. This part should be drastically improved otherwise it is not possible to     understand the lack of specific studies in the literature on the issue introduced by the authors     nor the added value of this research.

The methodology used in this work, is not new. What is novel on this?

I cannot see any general conclusions of this work. Again, what is the advance in respect of our similar papers?

Maps used in this paper are cartographically poor. Author needs to correct them.

Author Response

(The authors gave the same response as above.)

Round 2

Reviewer 3 Report

The authors have incorporated all the comments, and the manuscript may be accepted in its present form.